# Text mining-based word representations for biomedical data analysis and protein-protein interaction networks in machine learning tasks

**Halima Alachram** [1] *, **Hryhorii Chereda** [1], **Tim Beißbarth** [1], **Edgar Wingender** [2], **Philip Stegmaier** [2]

**1** Department of Medical Bioinformatics, University Medical Center, Göttingen, Lower Saxony, Germany,
**2** geneXplain GmbH, Wolfenbüttel, Lower Saxony, Germany

* halima.alachram@bioinf.med.uni-goettingen.de

**Data Availability Statement:** All relevant data are within the paper and its Supporting Information files.

## Abstract

Biomedical and life science literature is an essential way to publish experimental results. With the rapid growth of the number of new publications, the amount of scientific knowledge represented in free text is increasing remarkably. There has been much interest in developing techniques that can extract this knowledge and make it accessible to aid scientists in discovering new relationships between biological entities and answering biological questions. Making use of the *word2vec* approach, we generated word vector representations based on a corpus consisting of over 16 million PubMed abstracts. We developed a text mining pipeline to produce word2vec embeddings with different properties and performed validation experiments to assess their utility for biomedical analysis. An important pre-processing step consisted in the substitution of synonymous terms by their preferred terms in biomedical databases. Furthermore, we extracted gene-gene networks from two embedding versions and used them as prior knowledge to train Graph-Convolutional Neural Networks (CNNs) on large breast cancer gene expression data and on other cancer datasets. Performances of resulting models were compared to Graph-CNNs trained with protein-protein interaction (PPI) networks or with networks derived using other word embedding algorithms. We also assessed the effect of corpus size on the variability of word representations. Finally, we created a web service with a graphical and a RESTful interface to extract and explore relations between biomedical terms using annotated embeddings. Comparisons to biological databases showed that relations between entities such as known PPIs, signaling pathways and cellular functions, or narrower disease ontology groups correlated with higher cosine similarity. Graph-CNNs trained with word2vec-embedding-derived networks performed sufficiently good for the metastatic event prediction tasks compared to other networks. Such performance was good enough to validate the utility of our generated word embeddings in constructing biological networks. Word representations as produced by text mining algorithms like word2vec, therefore are able to capture biologically meaningful relations between entities. Our generated embeddings are publicly available at https://github.com/genexplain/Word2vec-based-Networks/blob/main/README.md.

**Funding:** This work was supported by the German Ministry of Education and Research (Bundesministerium für Bildung und Forschung, BMBF) project iDDSEM MyPathSem (ID 031L0024A+B). The funders had no role in study design, data collection and analysis, decision to publish, or preparation of the manuscript.

**Competing interests:** The authors have declared that no competing interests exist. The commercial affiliation 'geneXplain GmbH' did not play any role in the study and it does not alter our adherence to PLOS ONE policies on sharing data and materials. The funder provided support in the form of salaries for authors HA, HC, PS, TB and EW, but did not have any additional role in the study design, data collection and analysis, decision to publish, or preparation of the manuscript. The specific roles of these authors are articulated in the 'author contributions' section.

# 1 Introduction

The field of Natural Language Processing (NLP) is concerned with the development of methods and algorithms to computationally analyze and process human natural language. Solutions in this domain often have practical significance for everyday applications such as conversion between written and spoken language to enhance media accessibility, translation between linguae, optical character recognition (OCR) for street sign detection in traffic assistance systems, or document/media content classification for recommendation systems. In biomedical research, NLP is of importance to extract reported findings, e.g., protein-protein interactions, from scientific texts. Several studies employed supervised machine learning algorithms to identify and extract knowledge from scientific literature [1–4], which requires extensive manually labeled datasets for training.

A novel approach was recently introduced that applied neural networks (NNs) to learn high-dimensional vector representations of words in a text corpus that preserve their syntactic and semantic relationships [5]. The word2vec method, proposed by Mikolov et al. [5], embedded words in a vector space by predicting their co-occurrence so that words with similar meaning had a similar numerical representation. Word2vec can effectively cluster similar words together and predict semantic relationships. As produced by word2vec, word embedding allows computing relations between words obtained from a large unlabeled corpus, e.g., using their vector cosine similarity.

Several works in the biomedical research field have since adopted word vector presentations for various tasks like named entity recognition (NER) [6, 7], medical synonym extraction [8], as well as extraction of chemical-disease relations [9], drug-drug interactions [10] or protein-protein interactions [11]. Many studies have used PubMed abstracts [12], citations, or full text articles as a standard to generate word embeddings. However, usually a study examines a specific analysis task for defined aims. Each study uses input corpora including PubMed to achieve certain aims by considering different strategies such as adding a domain knowledge to obtain specific embeddings or by applying different evaluation methods. PubMed [12] is always the best library used to evaluate different word embedding strategies due to the large and valuable biomedical knowledge included. The novelty of such methods usually lies in the techniques used for the corpus processing and/or the methods used to evaluate and validate their utility in downstream analysis. Therefore, the generated embeddings are assessed for their quality by considering particular pre-processing procedures or by using different size of the input corpus as well as by applying different evaluation and validation techniques. Different validation strategies have been proposed to assess the quality of word embeddings. Wang et al. evaluated word embeddings' performance generated from four different corpora, namely clinical notes, biomedical literature (articles from PubMed Central (PMC)), Wikipedia, and news articles [13]. The evaluation was performed qualitatively and quantitatively. Their experimental results showed that embeddings trained on clinical notes are closer to human judgments of word similarity. They also demonstrated that word embeddings trained on general domain corpora are not substantially inferior in performance than those trained on biomedical or clinical domain documents. Zhang et al. assessed both the validity and utility of biomedical word embeddings generated using a sub-word information technique that combines unlabeled biomedical literature from PubMed with domain knowledge in Medical Subject Headings (MeSH) [14]. They evaluated the effectiveness of their word embeddings in BioNLP tasks, namely a sentence pair similarity task performed on clinical texts and biomedical relation extraction tasks. Their word embeddings have led to a better performance than the state-of-the-art word embeddings in all BioNLP tasks.

Several recent studies, particularly in molecular biology, have also considered word embeddings to represent biomedical entities and their functional relationships. Du et al. trained a gene embedding from human genes using gene co-expression patterns in data sets from the GEO databases and achieved an area under the curve score (AUC) of 0.72 in a gene-gene interaction prediction task [15]. Chen et al. employed NER tools to recognize and normalize biomedical concepts in a corpus consisting of PubMed abstracts. They trained four concept embeddings on the normalized corpus using different machine learning models. They assessed the concept embeddings' performance in both intrinsic evaluations on drug-gene interactions and gene-gene interactions and extrinsic evaluations on protein-protein interaction prediction and drug-drug interaction extraction. Their concept embeddings achieved better performance than existing methods in all tasks [16].

In other studies, word embeddings were used as input features to improve machine learning algorithms' performance. Kilimci et al. used different document representations, including term frequency-inverse document frequency (TF-IDF) weighted document-term matrix, mean of word embeddings, and TF-IDF weighted document matrix enhanced with the addition of mean vectors of word embeddings as features. They analyzed the classification accuracy of the different document representations by employing an ensemble of classifiers on eight different datasets. They demonstrated that the use of word embeddings improved the classification performance of texts [17].

Evaluating the validity of word embeddings by examining semantic relations can improve the transparency of word embeddings and facilitate the interpretation of the downstream applications using them. In many studies, word2vec has been applied to PubMed abstracts as a state-of-the-art method to generate word embeddings. However, their performance can differ significantly given different tasks for evaluation and validation. In this study, we applied word2vec to generate biomedical word embeddings using a corpus consisting of over 16 million PubMed abstracts and thoroughly validated their ability to capture biologically meaningful relations. Our generated word embeddings differ in the preprocessing phase which included a new procedure and the methods used to evaluate their performance for biomedical analysis. The new pre-processing procedure consists of substituting synonymous terms of biomedical entities by their preferred terms. Our validation methods are similar to those used by Chen et al. [16]. However, their evaluations concentrated on genes by considering drug-gene and gene-gene interactions. Our assessment covers vector cosine similarity of relations in protein-protein interactions (PPIs), common pathways and cellular functions, or narrower disease ontology groups using existing knowledge in biomedical databases.

Most word embeddings have been trained using either the word2vec [5] or the GloVe [18] model, which uses information about each word's co-occurrence with its nearby words to represent it in a distinct vector. Word2vec employs negative sampling and sub-sampling techniques to reduce the computational complexity. Word embeddings, learned by word2vec or other methods such as Skip-Gram or Continuous Bag-of-Words that predict a word's context from raw text using a target word, are called static embeddings. Such static embeddings are useful for solving lexical semantic tasks, particularly word similarity and word analogy, and representing inputs in downstream tasks [19]. More recent techniques in language modeling were unsupervised pre-trained language models such as BERT (Bidirectional Encoder Representations from Transformers) [20] and ELMO (Embeddings from Language Models) [21] that create contextualized word representations. Such models support fine-tuning on specific tasks and have shown effective performance improvements in diverse NLP tasks such as question answering and text classification. BioBERT [22] (Bidirectional Encoder Representations from Transformers for Biomedical Text Mining) was initialized with the BERT model to pre-train domain-specific language representations on large-scale biomedical articles. BERT

outperformed the NLP state-of-the-art on a variety of biomedical and clinical tasks. However, training BERT is computationally expensive due to its high model complexity and a large amount of training data needed to achieve acceptable accuracy.

Moreover, convolutional neural networks (CNNs) are among the often applied deep-learning network architecture that have delivered a good performance in image recognition and classification. CNN models were also effective for NLP tasks such as text classification [23, 24]. They have been used in bioinformatics [25], namely in drug discovery and genomics [26], and motivated further progress on graph structured prior information with promising results on the prediction of metastatic events [27, 28]. Graph Convolutional Neural Networks (Graph-CNNs) have been proven to be effective in capturing structural information in graphs [29]. Graph-CNNs have been effectively used for a variety of NLP applications such as machine translation [30]. Information from word embedding can be represented as edges in a graph with words as nodes. Thereby, Graph-CNNs are an effective way of representing such graphs and validating the embedding utility. Our word embeddings were assessed for their biological utility on a metastatic event prediction task. We trained Graph-CNNs on cancer gene expression data with gene-gene networks derived from word2vec embedding as prior knowledge to predict the occurrence of metastatic events. Graph-CNNs achieved comparable performance with word2vec-embedding-derived networks on liver, prostate and lung cancer data, and a slightly better performance on breast cancer data compared to protein-protein interaction networks or networks derived using other word embedding algorithms.

## 2 Materials and methods

Word embeddings generated for this study were based on a corpus of 16,558,093 article abstracts from the public PubMed FTP repository.

### 2.1. Corpus pre-processing and training of word embeddings

Natural language text data usually require an amount of preparation before they can be fed into the model training. For the purpose of pre-processing of the text corpus we implemented a pipeline that conducted the steps depicted in Fig 1. The first phase applied classical processing steps such as lowercasing, lemmatization, and removal of punctuation and numerical forms. We assumed replacing synonyms of biomedical terms with their main terms can affect the similarity between words in a way to better capture functional relationships between biomedical entities. Thus, we introduced an optional step before starting training, in which synonymous terms were substituted by externally defined main terms. This corpus was then used to train the word2vec model. For training we used the word2vec implementation in Gensim

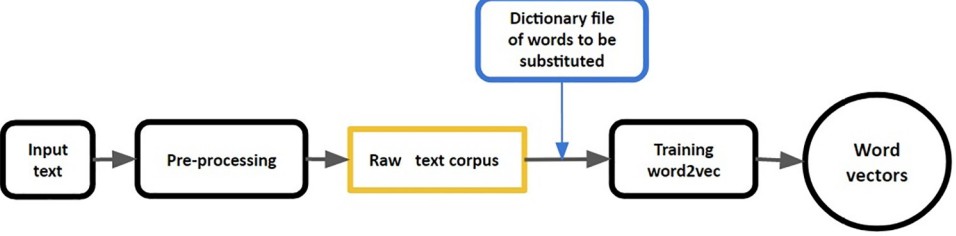

**Fig 1. Applied workflow to generate word2vec embeddings.** The process of generating word embeddings consists of two phases: A preprocessing and a training phase. Before training, an additional step conducts replacing synonymous terms with their main terms form biomedical databases.

[31] with context window of size 5, minimum count is equal to 5 and 300 neurons which is also the number of generated vector dimensions.

For this study, we generated the following two word2vec embeddings: 'Embedding_v1' in which synonymous terms of genes, diseases, and drugs were substituted by their preferred terms from HUGO [32], MeSH [33], and DrugBank [34], respectively; and 'Embedding_v2' for which the same preprocessing strategies and the same training process were applied but without replacing synonyms. Moreover, we assigned type labels using the same biomedical databases mentioned above in order to filter similarities. This enabled us to compare similarities between entities in obtained embeddings to existing knowledge in biomedical databases.

## 2.2. Validation of word embeddings

To provide an expedient tool for biological research, relative locations of terms within the vector space of the embedding should exhibit agreement with existing biological knowledge. Our validation addressed protein-protein interactions, signaling pathways and biological processes, drug targets and human diseases, which have been and continue to be of interest in many biomedical research projects. The conducted validation experiments therefore examined whether vectors of members within groups defined by respective biological databases featured increased cosine similarities compared to randomly sampled entities.

**2.2.1. Signaling pathways, biological processes and human diseases.**   Reactome 72 [35] and TRANSPATH® 2020.2 [36] pathway-gene assignments as well as Gene Ontology (GO, release 2020-03-25) [37] biological process-gene assignments were extracted from the geneXplain platform [38] version 6.0. Disease terms covered by the embedding were mapped to 139 groups of the Human Disease Ontology version 2019-05-13 [39] with more than 5 and less than 1000 member diseases. We calculated medians, lower and upper quartiles of cosine similarities for gene pairs within pathways and biological processes with at least 10 and not more than 3000 genes as well as for disease pairs within the 139 disease groups. In addition, we calculated medians, lower and upper quartiles for 2000 randomly sampled gene pairs and for 700 randomly sampled disease pairs that were not contained in selected groups (S1–S4 Files).

**2.2.2. Protein-protein interactions.**   Known protein-protein interactions were extracted from Reactome 63 for 4254 genes (16727 interactions) with vector presentations in the embedding. For the purpose of comparison, we sampled 10,000 random gene pairs and the same number of gene pairs with known interactions (S5 File).

**2.2.3. Drug-gene associations.**   The DrugBank [34] database combines detailed drug information with comprehensive drug target information. We extracted genes associated with each drug reported in DrugBank with type target. By considering 5234 drugs and their target genes, we created drug pairs based on the common genes that two drugs share in each pair. Drug-gene associations were obtained from DrugBank release 4.5.0 and cosine similarities of 50000 drug pairs with at least one shared target gene were compared to 50000 drug pairs without common target genes. Moreover, to examine the variability of the similarity distribution of drug pairs based on the number of genes they share, we sampled three drug pair groups (group1: no genes, group2: < = 5 genes, group3: < = 9 genes) (S6 File).

## 2.3. Gene expression data

**2.3.1. Breast cancer data.**   Graph-CNNs were trained on a breast cancer data set compiled by Bayerlová et al. [40]. The data consisted of 10 microarray data sets measured on Affymetrix Human Genome HG-U133 Plus 2.0 and HG-U133A arrays which are publicly available from the Gene Expression Omnibus (GEO) [41] under accession numbers GSE25066, GSE20685, GSE19615, GSE17907, GSE16446, GSE17705, GSE2603, GSE11121, GSE7390, and GSE6532.

The RMA (robust multi-array average) probe-summary algorithm [42] was applied to normalize each data set separately after which they were combined and further normalized using quantile normalization applied over all datasets. When more than one probe was associated with a gene, the probe with the highest average expression value was chosen, leading to 12179 genes. Training set classes consisted of 393 patients with metastasis within the first 5 years and 576 patients without metastasis between 5 and 10 years after biopsy.

**2.3.2. Other types of cancer.** We further applied Graph-CNNs to classify normal vs liver, lung or prostate tumor tissue as well as to predict FOLFOX therapy sensitivity of colorectal cancers. Gene expression data of FOLFOX therapy responders and non responders were obtained from GEO series GSE28702 [43]. Data sets were compiled from GEO series GSE6222, GSE29721, GSE40873, GSE41804, GSE45436, GSE62232 for liver cancer, GSE10799, GSE18842, GSE19188 for lung cancer, and GSE3325, GSE17951, GSE46602, GSE55945 for prostate cancer. The expression measurements were normalized using the justGCRMA method of the R/Bioconductor package gcrma versions 2.60.0 (FOLFOX response data) and 2.56.0 (other cancer data sets) [44]. For the cancer data sets assembled from different GEO series, batch correction was carried out using the R/Bioconductor package limma, version 3.40.6, with batches corresponding to source GEO series accessions [45] (sample and batch information for the normal vs. cancer data is given in S8 File). Information to map probe set identifiers to human gene symbols was obtained from Ensembl version 102 [46] using the R/Bioconductor package biomaRt [47], resulting in about 8500 genes after intersecting the microarray genes with PPI networks described in the section 2.4.

## 2.4. PPI networks

A broad range of machine learning models have been developed to analyze high-throughput datasets in the aim of predicting gene interaction and identifying prognostic biological processes. Recently, biomedical research has shown the ability of deep learning models in learning arbitrarily complex relationships from heterogeneous data sets with existing integrated biological knowledge. This biological knowledge is often represented by interaction networks. The high data dimensionality and the complexity of biological interaction networks are significant analytical challenges for modeling of the underlying systems biology. In this section, we present the PPI networks derived from different sources and used as prior knowledge to structure gene expression data.

**2.4.1. Human protein reference database.** In a recent study, Chereda et al. [27] employed the Human Protein Reference Database (HPRD) protein-protein interaction (PPI) [48] network to structure gene expression data of breast cancer patients. Genes from gene-expression data were mapped to the vertices of the PPI network yielding an undirected graph with 7168 matched vertices consisting of 207 connected components. The main connected component had 6888 vertices, whereas the other 206 components each contained 1 to 4 vertices. Since the approach of utilizing prior network information in Graph CNNs required a connected graph [29] training was carried out on the gene set of the main connected component. In this study, we used the same PPI network with the same approach of Chereda et al. [27].

**2.4.2. Word2vec embedding-based networks.** We created two gene-gene networks (Embedding_net_v1 and Embedding_net_v2) from the embedding version that excluded synonyms (Embedding_net_v1) and from the other version where word synonyms were taken into account (Embedding_net_v2). Both networks consisted of gene pairs with edges weighted by their cosine similarity values. The cosine similarity threshold was set to 0.65 yielding the Embedding_net_v1 network with 10730 genes in 4397 connected components with a main component of 6092 vertices and the Embedding_net_v2 network with 10729 genes in 4399

components with a main component covering 6106 vertices. The main connected components of Embedding_net_v1 and Embedding_net_v2 networks shared 5750 genes, therefore overlapping in the majority of vertices.

**2.4.3. STRING-derived network.** The STRING database [49] is a collection of protein-protein associations which can be derived from one or more sources such as gene neighborhoods, gene co-occurrence, co-expression, experiments, databases, text-mining, and whose confidence is expressed by an aggregated score computed from scores of the individual interaction sources. We considered the text-mining score as well as the combined score of all the interaction sources to build weighted protein-protein interaction networks. This way, the classification performance of Graph CNNs trained on the STRING text-mining network could be compared to Graph CNNs with prior knowledge from word2vec embedding-based networks. Like with the HPRD PPI, we mapped the genes to the two constructed STRING networks and supplied their main components to the training process. Score thresholds were chosen to obtain comparable number of vertices as in the HPRD PPI.

**2.4.4. BERT-embedding-derived network.** BERT (Bidirectional Encoder Representations from Transformers) [20] is a recently contextualized word representation model. The main technical innovation of BERT is the use of bidirectional transformers. BERT was pretrained in English Wikipedia and Books Corpus as a general language representation model. BioBERT [22] is a language representation model based on BERT and designed for biomedical text mining tasks. It was initialized with the BERT model provided by Devlin et al. in 2019 [20] and pre-trained on PubMed abstracts and PubMed Central full-text articles (PMC). We used the pre-trained BioBERT weights of 'BioBERT-Base v1.0' that was trained using the same vocabulary of BERT base (12-layer, 768-hidden, 12-heads, 110M parameters) on English text and 200k PubMed abstracts in addition. We converted the pre-trained TensorFlow checkpoint model to Pytorch [50], extracted the numerical vectors of 768 dimensions each and calculated the cosine similarities between entities to eventually extract a gene-gene network. The number of proteins in the main connected component was also kept according to the comparable number of vertices in the HPRD PPI.

**2.4.5. Random network.** For further comparison, we created an unweighted, random network containing the same 6888 vertices that were mapped to HPRD PPI. Each vertex was connected to 8 randomly chosen vertices except itself. Repetitions of vertices were possible. As a result, the nodes' degrees form a unimodal distribution lying in the interval [8, 30] with a mean value of 15.991, median 16, and standard deviation 2.80. To compare the performance of GCNN depending on the underlying networks, this version of a random network was used to structure the breast cancer data in section 2.3.1

Additionally, we createad a random network with random weights, modifying the random unweigted network from previous paragraph by assigningd to each edge a random value from the interval [0.65, 1]. This random network with random weights was used for comparison as a prior knowledge for the GCNN trained on datasets described in the section 2.3.2.

## 2.5. Graph-convolutional neural network (CNN)

One of the approaches for validation of the embedding networks is to analyze how the underlying molecular network influences performance of the machine learning method utilizing prior knowledge. The Graph-CNN method was applied on the breast cancer dataset introduced in section 2.3 in a recent study [28]. Also, we used other cancer datasets described in section 2.3.2. For all the datasets, the machine learning task is to predict a binary endpoint for a patient. The schema of the prediction workflow can be found in [26, of Fig 1]. As it was applied in the study [27], we subtracted the minimal value of the data from each cell of the

quantile normalized gene expression matrix to keep the gene expression values non-negative. The classification accuracy of Graph-CNNs was compared for different sources of network prior information: HPRD, Embedding_net_v1, Embedding_net_v2, STRING and BioBERT-based network. As for embedding networks, we utilized weighted and unweighted (taking into account only topology) versions. The vertices were mapped to the genes of gene expression data and weighted edges were filtered according to a threshold value. We considered thresholds higher than 0.5 for cosine similarity between vertices and arrived at the values of 0.63 and 0.65 to keep the number of vertices mapped to the gene symbols comparable. The main connected component of the underlying graph was used to structure the data. The performance was assessed by 10-fold cross-validation. For each of the data splits the model was trained on 9-folds and the classification was evaluated using 10-fold as validation set. For each dataset, the architecture and hyperparameters of Graph-CNN remained the same for all underlying molecular networks. The Graph-CNNs were trained on the same number of epochs for each data split. For the majority of the prior knowledge networks, Graph-CNN was trained with 100 epochs, but for some versions of prior knowledge, a smaller number of epochs showed better results since convergence of gradient descent was happening faster. The most common evaluation metrics were used: area under curve (AUC), accuracy and F1-weighted score. The metrics were averaged over folds and the standard errors of their means were calculated. We compared performances based on weighted and unweighted networks.

## 2.6. Assessment of text corpus size effect

We tested how the text corpus size influences the variability of word representations and compared the similarities between given concepts obtained from four resulting embeddings. The embeddings were produced by applying word2vec to four text corpora of different sizes. The four text corpora were of size 4M, 8M, 12M, and ~>16M. We selected 10 terms of different entity types, the genes brca1, psen1 and egf, the medical terms breast neoplasms, eczema, sleep and schizophrenia, and the molecular compounds ranitidine, lactose, and cocaine. The selected terms are among the ones that tend to appear frequently in literature which makes them have strong reltionships with their neighbors. The similarity variance of these relationships would reveal the effect of the text corpus size. Besides, this would also demonstrate how biologically meaningful are those relationships. For each entity term, we calculated its first 10 nearest neighbors and selected the ones that are commonly present in the four resulting embeddings (S7 File).

## 3 Results

### 3.1. Validation results

To demonstrate the utility of our word2vec embeddings in data analytical applications, we examined the agreement of cosine similarities between words according to their vector representations with information extracted from biomedical knowledgebases (see Materials and methods). As a result, pairs of genes with known interactions in the Reactome database showed on average higher cosine similarities than gene pairs without known interaction in the same database (Fig 2). Similarly, cosine similarities of drugs with overlapping target gene sets were, on average, higher than similarities between drugs without common target genes. Furthermore, cosine similarities within Reactome and TRANSPATH® pathways, as well as within GO biological processes, were increased compared to median cosine similarities of randomly sampled gene pairs (Fig 2). Regression curves estimated for the medians revealed a correlation between the number of pathways or GO category members and the median similarity, with higher values for smaller gene sets. We think that gene pairs in smaller pathway networks or biological processes were more likely to correspond to direct molecular interactors that

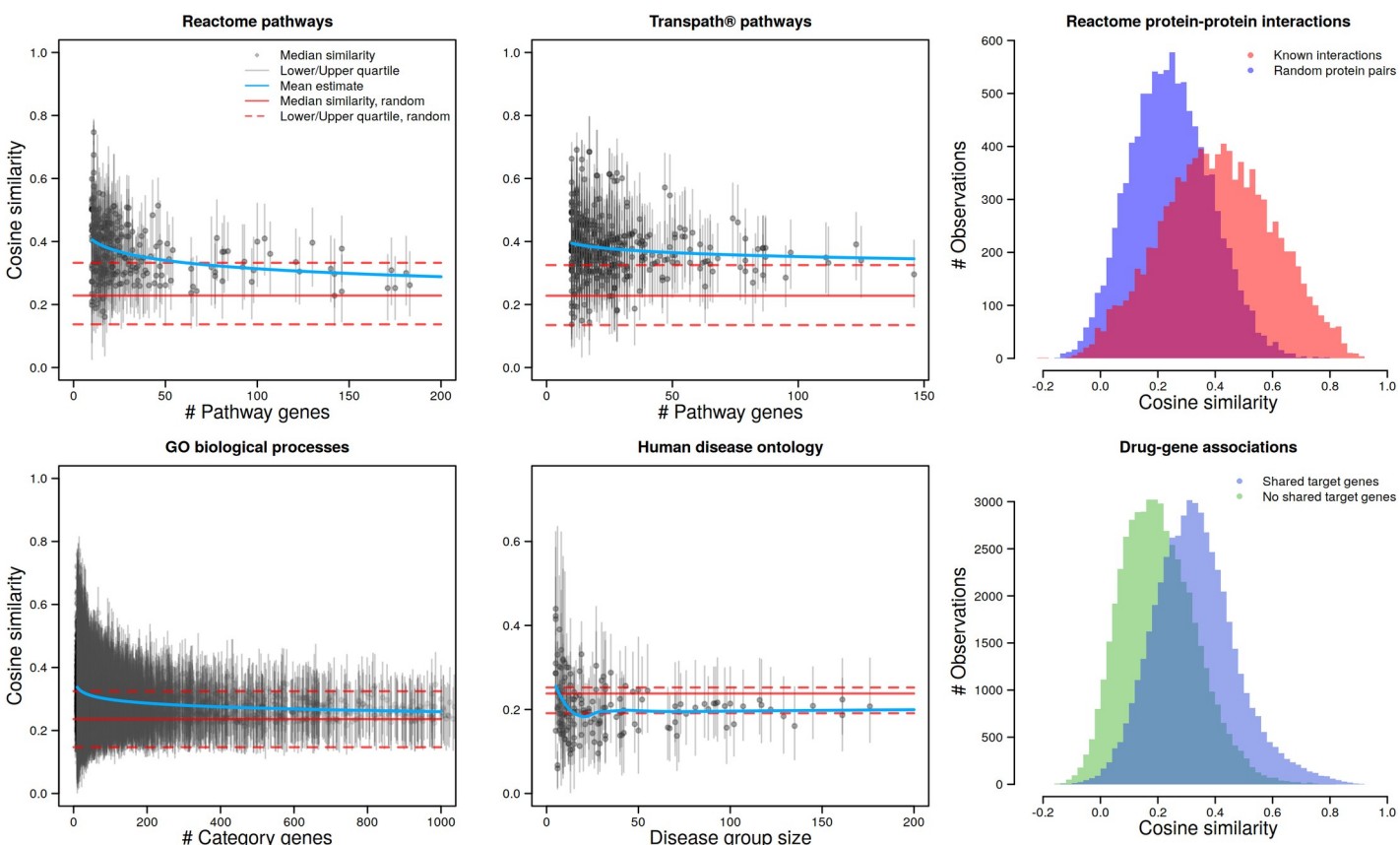

**Fig 2. Validation of the word2vec embedding with existing knowledge from biomedical resources.** Reactome pathways, TRANSPATH® pathways, GO biological processes and Human disease ontology present median cosine similarities as well as their lower and upper quartiles within groups of given number of members (genes or diseases, respectively) and for random samples (see Material and methods). Mean estimates were computed by fitting the decay function $f(x) = (x^a + b)^{-1}$ to medians, with the exception of the Human disease ontology comparison where a non-parametric local regression (Loess) was applied. Reactome protein-protein interactions and drug-gene associations show histograms of genes with or without known PPIs and of drugs with or without shared target genes, respectively.

share a close functional context than in pathway or functional categories with a higher number of members. The embedding, in many cases, indeed captured these relations. While disease-disease cosine similarities within Human Disease Ontology (HDO) groups also revealed such a trend for groups with less than 25 members, median similarities within groups were often smaller than for randomly chosen disease pairs (Fig 2). Therefore, disease-disease relations captured by broader HDO groups did not correspond well with vector presentations of the embedding. Better correspondence was observed for narrower disease groups but did not exceed similarities of random disease pairs. Full plots are provided as S1–S6 Figs. Moreover, drug-drug similarities were also assessed by the number of genes shared by two drugs. Median cosine similarities of drug pairs have increased as the number of shared genes increased (S7 Fig).

For comparison to the results of the word2vec-based embeddings, the same data sources were applied as for Fig 2 with embeddings obtained from Bio-BERT. Correlations with biological knowledge didn't show enough reliability for comparison and validation with Bio-BERT-based similarities as depicted in S8 Fig.

## 3.2. Graph-CNNs results

**3.2.1. Brest cancer data.** Graph-CNN models were trained on the breast cancer data from section 2.3.1 to predict an occurrence of a metastatic event, utilizing different prior knowledge.

**Table 1. The results of how weighted underlying networks influence the performance of Graph CNNs trained on gene expression data.**

| Network | Vertices | Similarity Threshold | AUC [%] | Accuracy [%] | F1-weighted [%] | Epochs |
|---|---|---|---|---|---|---|
| Embedding_net_v1 (weighted) | 6092 | 0.65 | **83.09±0.97** | **76.67±1.14** | **76.45±1.14** | 100 |
| Embedding_net_v2 (weighted) | 6086 | 0.68932 | 82.05±1.08 | 75.03±0.70 | 74.74±0.76 | 100 |
| Embedding_net_v1 (weighted) | 6775 | 0.63 | 82.53±1.46 | 75.62±1.72 | 75.29±1.73 | 100 |
| Embedding_net_v2 (weighted) | 6774 | 0.6774 | 81.87±1.39 | 75.33±1.22 | 75.16±1.28 | 40 |
| STRING (text mining weighted) | 6840 | 0.744 | 81.76±1.95 | 75.97±1.99 | 75.63±2.02 | 100 |

The networks were compared based on the similarity threshold and number of vertices included. 'Vertices' are the vertices in the main connected component. 'Similarity Threshold' is the minimum weight value to keep connections between genes. AUC, Accuracy, and F1 weighted are the evaluation metrics used in the scopes of 10-fold cross validation. 'Epochs' is the number of passing the entire dataset through the neural network.

The models with gene-gene networks derived from the embeddings showed best performance compared to HPRD PPI, STRING-derived networks, BioBERT-derived network, and random network, in classifying patients into two groups, metastatic and non-metastatic. The networks were compared based on the similarity threshold and the number of vertices included. The architecture of Graph-CNN consists of 2 convolutional layers. Two convolutional layers were used with 32 convolutional filters. Maximum pooling of size 2 applies to both of the convolutional layers. Two fully connected layers have 512 and 128 nodes, consequently.

We utilized 10-fold cross validation to assess the performance of Graph-CNN models, as stated in the section 2.5. Table 1 presents the performance of Graph-CNNs trained with the word2vec-embedding networks (Embedding_net_v1 and Embedding_net_v2) and STRING derived network incorporating the edge weights. For STRING, the edge weights are the scores computed based on text-mining techniques (see Materials and methods). We didn't consider a weighted BioBERT-derived network since the minimal weight was already 0.938 for edges forming a network with around 6000 vertices, which is not that much different from 1.0. We can see that Embedding_net_v1 demonstrated a better performance than Embedding_net_v2 for almost the same number of vertices and better than the text-mining-based STRING network.

In Table 2, we compared how unweighted network topologies influence the classifier's performance depending on the similarity threshold and the number of vertices included. The baseline performance corresponds to HPRD PPI prior knowledge. STRING (combined) and

**Table 2. Influence of unweighted underlying networks on the performance of Graph CNNs trained gene expression data.**

| Network | Vertices | Similarity Threshold | AUC [%] | Accuracy [%] | F1-weighted [%] | Epochs |
|---|---|---|---|---|---|---|
| HPRD | 6888 | - | 82.57±1.25 | 76.07±1.30 | 75.82±1.33 | 100 |
| Embedding_net_v1 | 6092 | 0.65 | **83.02±1.09** | **76.38±1.44** | **76.14±1.47** | **40** |
| Embedding_net_v1 | 6775 | 0.63 | 82.41±1.20 | 76.03±1.53 | 75.69±1.53 | 100 |
| Embedding_net_v2 | 6874 | 0.675 | 82.37±1.36 | 75.04±1.25 | 74.84±1.17 | 40 |
| STRING (text mining) | 6840 | 0.744 | 81.67±2.01 | 76.07±1.50 | 75.61±1.57 | 25 |
| STRING (combined) | 6862 | 0.938 | 81.77±1.17 | 74.62±1.56 | 74.33±1.64 | 100 |
| BioBERT-based network | 6865 | 0.95245 | 82.26±1.27 | 74.81±1.53 | 74.61±1.50 | 40 |
| Random network | 6888 | - | 81.89±0.109 | 75.65±0.99 | 75.43±0.99 | 40 |

The networks were compared based on the similarity threshold and number of vertices included. 'Vertices' are the vertices in the main connected component. 'Similarity Threshold' is the minimum weight value to keep connections between genes. AUC, Accuracy, and F1 weighted are the evaluation metrics used. 'Epochs' is the number of passes the entire dataset through the neural network.

**Table 3. Influence of underlying networks on the performance of Graph CNNs trained gene expression data.** Liver cancer (347 patients). Binary classification Normal vs Tumor Tissue.

| Algorithm | Network | Vertices (Features) | Similarity Threshold | AUC [%] | Accuracy [%] | F1-weighted [%] |
|---|---|---|---|---|---|---|
| GCNN | HPRD | 8454 | - | 94.32±1.70 | 93.67±1.03 | 93.63±1.04 |
| GCNN | Embedding_net_v1 | 8524 | 0.65 | **94.73±1.69** | **93.66±1.11** | **93.64±1.11** |
| GCNN | Embedding_net_v1 (permuted vertices) | 8524 | 0.65 | 94.47±1.85 | 93.39±1.27 | 93.39±1.27 |
| GCNN | Random_net_random_weights | 8524 | 0.65 | 95.60±1.54 | 93.97±1.16 | 93.96±1.16 |
| Random Forest | - | 8524 | - | 97.62±0.95 | 96.84±0.90 | 96.83±0.90 |

BioBERT-based networks were considered only as unweighted since the weight thresholds to reach a comparable number of vertices were close to 1, 0.938 and 0.952, respectively.

The embedding networks have a threshold value allowing to change the strength of similarity between vertices. Change of threshold for Embedding_net_v1 from 0.63 to 0.65 increased the classification result in weighted and unweighted cases. We can also observe that for embedding networks, the incorporation of weight's edges increased slightly, although not substantially, the classification performance. Meanwhile, STRING and BioBERT-based networks do not bring any improvements compared to HPRD PPI or random network. Thus, Graph-CNNs showed the best results on our dataset, incorporating weighted Embedding_net_v1 with a threshold of 0.65.

**3.2.2. Other cancer data sets.** For the purpose of comparison, we have also trained Graph-CNNs on the other cancer type datasets described in the section 2.3.2. As in the previous section, we have tried several underlying networks as prior knowledge to estimate their influence on the classification performance measured using 10-fold cross-validation. We used HPRD PPI, Embedding_net_v1, and two versions of randomized networks. The first version had the same topology as the Embedding_net v1 but with permuted vertices. We intended to remove the biological information about the direct interactions of PPIs while preserving the topology. The second version was created as described in the second paragraph of section 2.4.5. The grid search of Graph-CNNs hyperparameters was used to optimize its architecture on each of the datasets. The architecture of Graph-CNN remained the same within one dataset for different networks of the prior knowledge. Tables 3–6 demonstrate the metrics values as performance estimates. Interestingly enough, we did not notice any substantial differences between HPRD PPI, Embedding_net_v1, and Embedding_net_v1 with permuted vertices. It is noticeable that with those prior knowledge networks, Graph-CNNs performs comparably to Random Forest (Tables 4–6), except to the liver cancer dataset, where Random Forest outperformed Graph-CNNs. Only in Tables 5 and 6 does the random network with random edges worsens the classification rates. In those cases, the Graph-CNN had convergence issues during the training.

**Table 4. Influence of underlying networks on the performance of Graph CNNs trained gene expression data.** Lung cancer (266 patients). Binary classification Normal vs Tumor Tissue.

| Algorithm | Network | Vertices (Features) | Similarity Threshold | AUC [%] | Accuracy [%] | F1-weighted [%] |
|---|---|---|---|---|---|---|
| GCNN | HPRD | 8454 | - | 95.62±1.35 | 93.62±1.25 | 93.70±1.25 |
| GCNN | Embedding_net_v1 | 8524 | 0.65 | **96.10±1.06** | **93.25±1.22** | **93.32±1.20** |
| GCNN | Embedding_net_v1 (permuted vertices) | 8524 | 0.65 | 94.55±1.62 | 92.51±1.24 | 92.59±1.22 |
| GCNN | Random_net_random_weights | 8524 | 0.65 | 95.60±1.27 | 91.74±1.36 | 91.82±1.33 |
| Random Forest | - | 8524 | - | 97.46±0.66 | 92.88±1.17 | 92.94±1.15 |

**Table 5. Influence of underlying networks on the performance of Graph CNNs trained gene expression data.** Prostate cancer (234 patients). Binary classification Normal vs Tumor Tissue. The GCNN was not able to converge on some folds utilizing a random network with random weights.

| Algorithm | Network | Vertices (Features) | Similarity Threshold | AUC [%] | Accuracy [%] | F1-weighted [%] |
|---|---|---|---|---|---|---|
| GCNN | HPRD | 8454 | - | 98.28±0.66 | 93.97±1.49 | 93.93±1.49 |
| GCNN | Embedding_net_v1 | 8524 | 0.65 | **97.84±0.81** | **92.72±1.43** | **92.68±1.44** |
| GCNN | Embedding_net_v1 (permuted vertices) | 8524 | 0.65 | 98.67±0.60 | 94.00±1.15 | 93.96±1.16 |
| GCNN | Random_net_random_weights | 8524 | 0.65 | 69.86±8.11 | 70.25±7.57 | 60.38±10.24 |
| Random Forest | - | 8524 | - | 98.00±0.77 | 92.75±1.42 | 92.72±1.43 |

### 3.3. Effect of text corpus size

We generated four embeddings trained with corpora of different sizes to examine the variability of cosine similarity values depending on the amount of training data. Fig 3 illustrates the results for selected terms of different types and their nearest neighbors. The cosine similarities are varying between the terms. For example, the similarity value between the "breast neoplams" and "ovarian neoplasms" had increased slightly as follows: 0.851 (4M), 0.855 (8M), 0.861 (12M), 0.862 (~16M). Breast neoplasm is one of the most frequent diagnosed neoplasms reported in biomedical literature. Many studies have also reported similarities between breast and ovarian cancer since they share similar mutations (tumor suppressors). On the other hand, the similarity between "brca1" and "brca2" is almost the same in the four embeddings (0.898, 0.893, 0.891, 0.898 for 4M, 8M, 12M, and ~16M, respectively) with a very high similarity compared to the other nearest neighbors of "brca1". BRCA1 and BRCA2 genes are the most common genes defined in literature with certain mutations and lead to an increased risk of breast and ovarian neoplasms. Similarly, for "schizophrenia" and "bipolar_disorder", their similarity has changed slightly (0.822, 0.825, 0.828, 0.829 for 4M, 8M, 12M, and ~16M, respectively). An overlap between schizophrenia and bipolar disorder has been commonly reported in the literature.

In contrast, the similarity between "eczema" and "atopy" had changed differently. It had decreased from 0.713 in the embedding with the corpus of size 4M to 0.669 in the one with 8M, to continue increasing again to 0.691 (12M) and 0.701 (~16M) while staying lower than their similarity in the embedding pre-trained with the smallest corpus.

Overall, we observed that the nearest neighbors of selected terms were assigned similarity values as well as a similar ranking that were varying slightly in the four embeddings for the majority of the selected terms. However, for common terms such as breast neoplasms, BRCA1, and schizophrenia and their nearest neighbors with which they tend to appear more frequently in biomedical literature, the similarity was more robust.

## 4 Discussion

In this paper, we focused on demonstrating the utility of word embedding-derived knowledge in uncovering valuable biological relationships and its application in machine learning tasks.

**Table 6. Influence of underlying networks on the performance of Graph-CNNs trained gene expression data.** Colorectal cancer, GSE28702_CRC_FOLFOX_responders (82 patients). Binary classification Responder vs Non-Responder.

| Algorithm | Network | Vertices (Features) | Similarity Threshold | AUC [%] | Accuracy [%] | F1-weighted [%] |
|---|---|---|---|---|---|---|
| GCNN | HPRD | 8454 | - | 73.25±5.18 | 76.67±3.98 | 75.61±4.26 |
| GCNN | Embedding_net_v1 | 8524 | 0.65 | **71.62±5.38** | **67.08±5.25** | **66.44±5.31** |
| GCNN | Embedding_net_v1 (permuted vertices) | 8524 | 0.65 | 71.38±4.27 | 67.92±4.76 | 67.72±4.76 |
| GCNN | Random_net_random_weights | 8524 | 0.65 | 63.38±6.26 | 59.44±2.95 | 56.22±3.99 |
| Random Forest | - | 8524 | - | 75.88±5.37 | 71.67±2.82 | 71.21±2.87 |

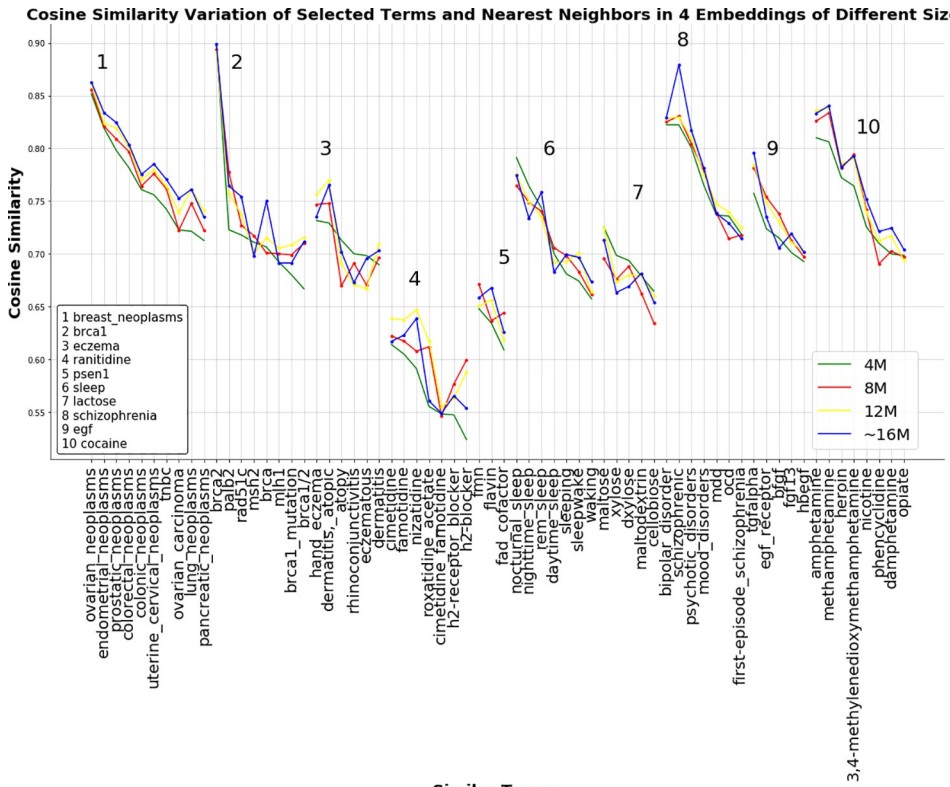

**Fig 3. Assessment of similarities between selected terms and their nearest neighbors present in 4 embeddings.** The selected terms are the genes brca1, psen1 and egf, the medical terms breast neoplasms, eczema, sleep and schizophrenia, and the molecular compounds ranitidine, lactose, and cocaine; and their nearest neighbors present in 4 embeddings trained on a text corpus of different sizes. Each numbered subplot represents the cosine similarities between a selected term and its nearest neighbors from the four embeddings. Each plotted line represents one embedding and each dot on a line is the similarity between a selected term and one nearest neighbor. A point on the line is the similarity value between a selected term and one nearest neighbor. Nearest neighbors are arranged according to decreasing cosine similarity for the 4M corpus.

While the text corpus used in this study consisted entirely of PubMed abstracts, our approach can be applied to complete scientific or other types of natural language texts. Our word embedding generation differs from the usual applications by incorporating an uncommon pre-processing step in addition to the conventional techniques, which is the substitution of biological entity synonymous terms by their preferred terms. The substitution was applied on gene, disease, and drug terms using existing knowledge in biomedical databases. By this procedure, we aimed to capture all contexts of the same concepts present as synonyms and to boost their neighborhood. Thereby, the model treats the different terms as one term and generates only one numerical vector. Such technique would affect the neighborhood of a word by normalizing the different variants of a term by mapping them to a single form. Therefore, such a normalization would help to reduce variability and, in some cases, ambiguity. We performed a computational analysis to validate similarities between biomedical entities, namely, genes, diseases, and drugs, using existing knowledge in biomedical databases. Comparisons showed that relations between known PPIs, common pathways and cellular functions, or narrower disease ontology groups correlated with higher vector cosine similarity. Gene pairs with known PPIs in Reactome have generally shown higher cosine similarities. Gene embeddings seem to be rich with semantic information about gene function. On the other hand, gene pairs with high

cosine similarities shown without known interactions in Reactome would lead to new investigations to uncover hidden functional relationships. Besides, gene pairs sharing common pathways in Reactome and TRANSPATH®, as well as common biological processes in GO, showed increased cosine similarities compared to the median of randomly sampled gene pairs. Similarities were also increased with smaller group sizes, which more likely represent direct molecular interactions. Disease pairs also showed increased cosine similarities within smaller human disease ontology (HDO) terms/groups, e.g., $< = 20$ diseases, which probably represent more specific disease classes. However, disease embeddings did not correspond well on the basis of median random similarity. This is potentially interesting to investigate why semantic relations between diseases differ from the HDO, although it would suggest that the embedding could harbor new insights for disease ontologists. Moreover, in comparison with word2vec, we assessed Bio-BERT performance using the same resources for PPIs, common pathways and disease-disease relations. Similarities between biological entities did not show an evident agreement with biomedical knowledge, even though, median similarities were higher than with word2vec performance. The reason for this might be that these similarities correspond to other types of relationships or to different contexts which have been captured with word2vec.

Corpus size effect assessment showed that similarities between selected terms were substantially affected by the corpus size. In general, we noticed that the first nearest neighbor for most terms was not strongly influenced by the corpus size even though it was not changing proportionally with the corpus size. The highest and strongest similarities were observed between "breast_neoplasms" and "ovarian_neoplasms" as well as between "brca1" and "brca2". This might be justified by the fact that these terms are very common in the present literature and words in each pair tend to occur more frequently in the same context. This could validate the ability to extract meaningful functional relationships between biomedical terms.

Additionally, to demonstrate the utility of the embedding in machine learning tasks, we assumed that similarities between biological entities might help create networks of a specific type. The results of Graph-CNN on the breast cancer dataset showed that weighted and unweighted Embedding_net_v1 allowed to increase the classification performance to predict breast cancer's metastatic event achieving the highest F1 score, AUC, and accuracy. This has demonstrated the biological utility of the embedding as prior knowledge for prediction of a metastatic event in breast cancer. Moreover, the change of similarity threshold of edge weights from 0.63 to 0.65 has led to an increase in performance; this could be due to the fact that "weak genes" contained in the network were filtered out. Besides, Random network-based demonstrated lower performance, although it is still to be investigated how simulated networks with different degree distributions would influence the classification error rate. It was also shown that the model trained with the Embedding_net_v1 network had performed better than with the Embedding_net_v2. The former was produced from the embedding in which synonymous terms replaced with their main terms. Such a procedure has surely influenced the embedding information and, in particular, the semantic relations between terms. For example, considering the gene WNT4 and its nearest neighbor WNT7a, the cosine similarity between them has increased from 0.798 in Embedding_v2 to 0.811 in Embedding_v1, respectively. Although the similarity was slightly increased, this has led to changing WNT7a from being the third neighbor of WNT4 to becoming its first neighbor. Another example, the top nearest neighbors of the TP73 (Tumor Protein P73) gene included the following terms in Embedding_v2: 'cdkn2c, cdkn2b, dapk1, rassf1, lzts1, dlec1'. However, this neighborhood list has completely changed in Embedding_v1 as follows: 'tp53, np73, tap63, mdm2, deltanp73, ing1'. This list included more reasonable neighbors in terms of gene partners. According to STRING database, tp53 and mdm2 are of the top genes that have functional links with tp73 based on evidence from experiments, curated databases and text-mining. This change was enhanced by replacing p73

with tp73. Moreover, for 'schizophrenia', some of the nearest neighbors were normalized which has affected the similarity computed for their variants and the standardized forms between the two embedding versions. The neighbors 'mood disorder' and 'affective disorder' were replaced with 'mood disorders' in Embedding_v1, which has led to an increase of the similarity from 0.746 between 'schizophrenia' and 'mood disorder' and 0.728 between 'schizophrenia' and 'affective disorder' to 0.776 between 'schizophrenia' and 'mood disorders'. Similarly, the replacement of 'asthma' with 'autistic disorder' has increased the similarity from 0.686 between 'schizophrenia' and 'asthma' in Embedding_v2 to 0.713 between 'schizophrenia' and 'autistic disorder' in Embedding_v1.

Additionally, we investigated the influence of Embedding_v1 network on Graph-CNN performance using four data sets and classifying normal vs liver, lung or prostate tumor tissue as well as predicting FOLFOX therapy sensitivity of colorectal cancers. The difference in performance between Embedding_v1, HPRD PPI, and Embedding_v1 with permuted vertices was not substantial over the aforementioned datasets. We assume, that "small world properties" could be a reason for that—the majority of vertices' pairs can be connected through the path with no more than 7 hops. 7-hops neighborhood were used by convolutional filters of all the Graph-CNNs we used. Also, for the lung and liver cancer, fully random network demonstrated similar performance. We hypothesize, that in the case of these datasets, Graph-CNN was able to pick up patterns necessary for classification regardless of the vertex's connectivity or network information. A biological reason could stand behind this phenomenon. For instance, for lung and liver cancer data sets where more heterogeneous and expression correlations between genes did not coincide well with provided network topologies. We also noticed that the performance with the network with permuted vertices was always comparable. Only in prostate and colorectal cancer datasets, a random network with random weights worsens the classification performance. This fact is worth to be investigated further. Furthermore, predictions of Graph-CNN applied to the same gene expression data used in this study with the HPRD PPI were explained in a recent study and provided patient-specific subnetworks [28]. An interesting research question brought up by this study is whether patient-specific subnetwork genes predicted using an embedding-based gene-gene network would give different insights into the tumor biology of a patient than those predicted using PPI networks. This might also provide biological insights into the molecular interactions in the network and help to validate the biological utility of the embedding-based network information.

Broadly translated our findings indicate that the performance obtained by Graph-CNN is sufficiently good to judge the utility of word2vec-embedding in creating gene-gene networks for machine learning tasks. Since the obtained results with word2vec-embedding-based networks are comparable with other networks, this would be a clear proof of our concept.

Furthermore, the influence of embedding-based networks can also be examined by considering text-mining-based networks other than STRING and BioBERT. One could also derive networks from BioBERT using different hidden layers. For our BioBERT-derived network, the vectors of words were extracted from the last hidden layer. The BERT authors extracted vector combinations from different layers. They tested word-embedding strategies by feeding those vector combinations to a BiLSTM (bidirectional long short-term memory used on a named entity recognition task and observing the resulting F1 scores [20]. The concatenation of the last four layers produced the best results on this specific task. It is generally advisable to test different combinations in a particular application since results may vary.

Using word2vec-based embeddings to create biological networks would be advantageous compared to other network's resources due to its straightforward application. Databases that maintain manually curated PPI data needs to be always up-to-date which is an expensive task. As biomedical literature is a primary resource to extract PPI data, it would be useful that text-

mining-based methods could be easily applied to perform this task. Bio-BERT and other similar methods require extensive computational power and resources to achieve good performance. However, word2vec is easy-to-handle and computationally inexpensive. It is able to achieve good performance without necessarily using large input data. As we showed in our assessment in section 3.3, similarities between common biomedical terms have not been much affected by a smaller input corpus. This has demonstrated that similarities of real biological relationships are robust even with smaller input corpora.

Although our examination was based only on gene-gene relations, it can be however extended to cover other types of relations. This utility of the biomedical embedding can be an advantage to create other networks of different types of entities such as disease or drug networks. We have also developed a web service based on this work to explore biomedical concepts present in our generated embeddings. It can be accessed under the link: https://ebiomecon.genexplain.com/. The service facilitates accessing the embedding information, and it provides functions to explore similarities of biomedical concepts, including the possibility to extract network vertices. Our embedding networks were extracted in the form of vertices. The implementation and the functionality of the web service will be described in more depth in a separate publication (manuscript in preparation).

## 5 Conclusion

In this study, we leveraged a state-of-the-art text-mining tool to learn the semantics of biomedical concepts presented in biomedical literature and to demonstrate the utility of learned embeddings for biomedical research and analysis. Our learned embeddings have been validated in computational analyses addressing protein-protein interactions, signaling pathways and biological processes, drug targets, and human diseases and showed agreement with existing biological knowledge. The results demonstrated that vector representations of biomedical concepts as produced by word2vec can capture meaningful biological relationships between biomedical entities. We also showed that semantic relations extracted from vast literature could be applied as prior knowledge in machine learning tasks.

## Supporting information

**S1 Fig. Gene-gene cosine similarities within Reactome pathways of given number of genes.** Median, lower and upper quartiles are presented for a random sample of 2000 gene pairs. A mean trend was estimated using the function $f(x) = (xa + b)-1$.
(TIF)

**S2 Fig. Gene-gene cosine similarities within GO biological processes of given number of genes.** Median, lower and upper quartiles are presented for a random sample of 2000 gene pairs. A mean trend was estimated using the function $f(x) = (xa + b)-1$.
(TIF)

**S3 Fig. Gene-gene cosine similarities within TRANSPATH® pathways of given number of genes.** Median, lower and upper quartiles are presented for a random sample of 2000 gene pairs. A mean trend was estimated using the function $f(x) = (xa + b)-1$.
(TIF)

**S4 Fig. Disease-disease cosine similarities within human disease ontology groups of given number of diseases.** Median, lower and upper quartiles are presented for a random sample of 700 disease pairs. A mean trend was estimated by non-parametric local regression (Loess).
(TIF)

**S5 Fig. Histograms of genes with or without known Reactome protein-protein interactions.** Each group contained a random sample of 10000 pairs.
(TIF)

**S6 Fig. Histograms of drug-gene associations with or without shared target genes.** Each group contained a random sample of 50000 drug pairs.
(TIF)

**S7 Fig. Drug-drug cosine similarity distributions with shared genes of given number in DrugBank.** Drug-drug groups were estimated by counting the number of shared genes between two drugs presented in the embedding. Group1 (no genes: median = 0.192, lower quartile = 0.108, upper quartile = 0,286), group2 (genes $\leq$ 5: median = 0.318, lower quartile = 0.229, upper quartile = 0,411), group3 (genes $\leq$ 9: median = 0.396, lower quartile = 0.292, upper quartile = 0,516).
(TIF)

**S8 Fig. Validation of the BioBERT embedding with existing knowledge from biomedical resources.**
(TIF)

**S1 File. Summary of gene-gene cosine similarities within biological processes in Gene Ontology (GO).**
(CSV)

**S2 File. Summary of gene-gene cosine similarities within pathways in Reactome.**
(CSV)

**S3 File. Summary of gene-gene cosine similarities within pathways in TRANSPATH®.**
(CSV)

**S4 File. Summary of disease-disease cosine similarities within disease groups in Human Disease Ontology (HDO).**
(CSV)

**S5 File. Gene pairs with or without known protein-protein interactions in Reactome.**
(CSV)

**S6 File. Drug pairs with or without shared target genes.**
(XLSX)

**S7 File. Cosine similarities of selected terms and their nearest neighbors in four corpora of different sizes.**
(CSV)

**S8 File. A sample sheet for the cancer vs. normal data.**
(XLSX)

## Acknowledgments

We thank Kamilya Altynbekova from geneXplain GmbH for providing GEO series accessions and CEL files for the normal vs. cancer comparisons.

## Author Contributions

**Conceptualization:** Halima Alachram, Hryhorii Chereda, Tim Beißbarth, Edgar Wingender, Philip Stegmaier.

**Methodology:** Halima Alachram, Hryhorii Chereda, Philip Stegmaier.

**Supervision:** Tim Beißbarth, Edgar Wingender, Philip Stegmaier.

**Validation:** Halima Alachram, Hryhorii Chereda, Philip Stegmaier.

**Visualization:** Halima Alachram, Philip Stegmaier.

**Writing – original draft:** Halima Alachram, Hryhorii Chereda, Philip Stegmaier.

**Writing – review & editing:** Halima Alachram, Hryhorii Chereda, Tim Beißbarth, Edgar Wingender, Philip Stegmaier.

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
