## [Decision Letter · Decision Letter 0]

24 Mar 2021

PONE-D-20-38223

Text mining-based word representations for biomedical data analysis and machine learning tasks

PLOS ONE

Dear Dr. Alachram,

Thank you for submitting your manuscript to PLOS ONE. After careful consideration, we feel that it has merit but does not fully meet PLOS ONE’s publication criteria as it currently stands. Therefore, we invite you to submit a revised version of the manuscript that addresses the points raised during the review process.

We look forward to receiving your revised manuscript.

Kind regards,

Khanh N.Q. Le

Academic Editor

PLOS ONE

Journal Requirements:

We note that one or more of the authors are employed by a commercial company: geneXplain GmbH.

2.1. Please provide an amended Funding Statement declaring this commercial affiliation, as well as a statement regarding the Role of Funders in your study. If the funding organization did not play a role in the study design, data collection and analysis, decision to publish, or preparation of the manuscript and only provided financial support in the form of authors' salaries and/or research materials, please review your statements relating to the author contributions, and ensure you have specifically and accurately indicated the role(s) that these authors had in your study. You can update author roles in the Author Contributions section of the online submission form.

2.2. Please also provide an updated Competing Interests Statement declaring this commercial affiliation along with any other relevant declarations relating to employment, consultancy, patents, products in development, or marketed products, etc.  

Reviewers' comments:

Reviewer's Responses to Questions

**Comments to the Author**

1. Is the manuscript technically sound, and do the data support the conclusions?

Reviewer #1: Partly

Reviewer #2: Yes

Reviewer #3: Partly

2. Has the statistical analysis been performed appropriately and rigorously? 

Reviewer #1: No

Reviewer #2: Yes

Reviewer #3: No

3. Have the authors made all data underlying the findings in their manuscript fully available?

Reviewer #1: No

Reviewer #2: Yes

Reviewer #3: No

4. Is the manuscript presented in an intelligible fashion and written in standard English?

Reviewer #1: Yes

Reviewer #2: No

Reviewer #3: Yes

5. Review Comments to the Author

Reviewer #1: The paper proposed a word embedding trained on 16M Pubmed abstracts using word2vec approach, Then the embedding was used as prior knowledge to train a graph convolutional neutral networks. The performance of the GNN with word2vec embeding outperforms other the GNNs with other embeddings on the metastaci event prediction task.

1. Comparing to the content, the scope of the title is too large.

2. The article is hard to read, need to be polished

3. please also provide a sample code about how to read the embedding file.

4. train word embedding on pubmed by word2vec approach is not new. From paper "A survey of word embeddings for clinical text", we can easily find many existing papers have trained word embedding on pubmed by different embedding method.

5. the performance evaluation is not sufficient. The title is "for machine learning tasks", but the performance was only evaluated on a not well-known breast cancer dataset. And the paper didn't compare the performance with the existing State of the art methods on the same dataset.

6. More figures about the methdology part will be helpful for reader to understand the method.

Reviewer #2: In this manuscript, the authors introduced a text mining pipeline to produce word2vec embeddings for biological data. The proposed technique was evaluated on different tasks. The manuscript was well-written. All data and methods were introduced clearly.

I have the following comments to be addressed by the authors:

The performance of using Embedding_v1 is slightly better than Embedding_v2. The authors need to discuss the advantages of using the substitution of synonymous terms of genes, diseases, and drugs by their preferred terms.

The authors gave one example in the discussion section in which using Embedding_v1 outperforms Embedding_v2. I recommend the authors providing additional examples.

The main concern is comparing the performance of the proposed model with BioBERT in the tasks such as Reactome pathways, TRANSPATH pathways, GO biological processes, and Human disease ontology. The authors need to show the performance of BioBERT on these tasks.

Did the authors test using other similarities rather than cosine similarity?

How did the authors select the Similarity Threshold?

Reviewer #3: Review of manuscript “Text mining-based word representations for biomedical data analysis and machine learning tasks” by Alachram and co-workers.

I enjoyed reading this manuscript as it explained in some (but not too much) detail the process of using text mining with word2vec representation in biomedical text mining. The manuscript is logically structured, and I found most of my questions answered when reading further. However, I see at least two weaknesses with this work in its current state. Neither would disqualify the manuscript from publication, but at least this reviewer would like to see some more discussion on these limitations:

The improvement in the CNN classifier when based on the word2vec network over existing networks is very small, and not completely consistent. More worryingly or surprisingly perhaps, no network seems to perform significantly better than the random (unweighted) network. The differences seem to correspond to no more than one or two patients in the test set with 97 patients. No random weighted network (with random weights) was compared with. Why? It would also be interesting to see a comparison with other common (non-CNN) classifiers. Are those better or worse?

The networks are only compared on one breast cancer dataset. Though I admit we have also used this particular example, it contains some of the strongest gene-disease associations (and gene-gene with BRCA1 and BRCA2, and these with TP53) in the literature. It is therefore a bit of an extreme case. As the networks perform similarly, it would be interesting to see how they do on a wider range of examples.

Other comments:

Is the preprocessing step substituting synonymous terms by their preferred terms really trivial and error free? Quite a few gene and protein symbols/names are ambiguous, for example. Is the substitution global, or dependent on date of publication or context? What is the error rate of this substitution? For example, the official HGNC symbol for the serine dehydratase gene is “SDS”, but “SDS” in the literature more often refers to sodium dodecyl sulfate. This gene appears to have been removed from the networks, or at least I could not find it in the eBioMeCon services. The gene symbol for the amyloid beta precursor protein is “APP”, which also has another meaning, especially after lowercasing (though the list of nearest terms on the eBioMeCon looks reasonable for this gene). How often is an unambiguous term substituted for an ambiguous but preferred one in ‘Embedding_v1’? How often is an ambiguous term substituted for a non-synonymous one?

Line 259: What is meant by “the number of proteins in the main connected component was also kept according to the comparable number of vertices in the HPRD PPI”? Did the authors fix the number of proteins in the main connected component to that in the existing PPT, and if so how? By removing edges below the “similarity threshold” as shown in Table 1?

Line 295: How were these 10 terms chosen? Based on known strong associations? As mentioned above, BRCA1 is one of the more frequently appearing genes in the corpus, and having one of the strongest associations with a disease (breast cancer), so it is a bit of an extreme case (not saying that such are not useful or interesting, but it should at least be mentioned when introduced).

Table 1 - could not a random network, with random weights, be used for comparison? It would also be highly interesting to see the overlaps between the predictions. Are the misclassified cases more or less the same for all networks? The would definitely warrant a closer inspection of the gene expression dataset.

Line 474: should be brca2, not “brac2” (different gene, not strongly associated).

6. PLOS authors have the option to publish the peer review history of their article (what does this mean?). If published, this will include your full peer review and any attached files.

Reviewer #1: No

Reviewer #2: No

Reviewer #3: No

---

## [Author Response · Author response to Decision Letter 0]

31 Aug 2021

Dear Editors,

Thank you for your email dated 24 March 2021 enclosing the reviewers’ comments and for inviting us to submit a revised version of our manuscript. We appreciate your valuable comments. We have carefully considered the comments and have revised the manuscript accordingly. We hope that the manuscript, after careful revision, meets your high standards.

Based on the reviewer comments, we have incorporated the following changes into the manuscript:

• We changed the title of the manuscript to make it more precise.

• We provided a sample code to read an embedding through a GitHub repository.

• We applied the same validation analysis described in sections 2.2 and 3.1 using the same data resources with similarities obtained from Bio-BERT to show the performance of Bio-BERT.

• We applied our machine learning task using new additional data sets as well as with a state-of-the-art classifier (Random Forest) for comparison.

• Accordingly, we have included the sections 2.3.2 to describe the new data sets and 3.2.2 to analyse the new results.

• Finally, we have edited the content of the manuscript at different places to improve the text and to incorporate all the suggestions.

Our responses are given in a point-by-point manner below. All the modifications have been highlighted in the manuscript.

Reviewers' comments:

Reviewer #1: The paper proposed a word embedding trained on 16M Pubmed abstracts using word2vec approach, Then the embedding was used as prior knowledge to train a graph convolutional neutral networks. The performance of the GNN with word2vec embeding outperforms other the GNNs with other embeddings on the metastaci event prediction task.

1. Comparing to the content, the scope of the title is too large.

Thank you for pointing this out. We have edited the title to make it more specific as follows:

“Text mining-based word representations for biomedical data analysis and protein-protein interaction networks in machine learning tasks”

2. The article is hard to read, need to be polished

We have edited few sections to make the article easier to read and tried to improve the text at different places.

3. please also provide a sample code about how to read the embedding file.

We have provided a sample code to read an embedding file along with our generated embeddings in our GitHub repository which is available under the following link:

https://github.com/genexplain/Word2vec-based-Networks/blob/main/README.md

We have also added this information in the article (Abstract).

4. train word embedding on pubmed by word2vec approach is not new. From paper "A survey of word embeddings for clinical text", we can easily find many existing papers have trained word embedding on pubmed by different embedding method.

We agree that PubMed has been widely as a standard to train word embeddings. However, each paper uses input corpora including PubMed to achieve certain aims by considering different strategies such as adding a domain knowledge to obtain specific embeddings or by applying different evaluation methods. PubMed is always the best library used to evaluate different word embedding strategies due to the large and valuable biomedical knowledge included. The novelty of such methods usually lies in the techniques used for the corpus processing and/or the methods used to evaluate and validate their utility in downstream analysis. Our study differs from others by considering the procedure of substituting synonymous terms during pre-processing. With few examples mentioned in the discussion section, we have showed that this procedure has influenced the similarity and the neighbourhood of certain entity terms. In addition, our evaluation methods examined different aspects which showed the utility of the embedded knowledge for downstream tasks. Moreover, we used the embeddings to generate gene-gene networks and used them to structure gene expression data in new approach that could bring valuable insights in personalized medicine using Graph-CNN. Such approach is helpful to validate the utility of word embeddings in creating biological networks. We have edited our introduction by highlighting the use of PubMed in different studies (line 66). 

5. the performance evaluation is not sufficient. The title is "for machine learning tasks", but the performance was only evaluated on a not well-known breast cancer dataset. And the paper didn't compare the performance with the existing State of the art methods on the same dataset.

We agree, that wording “machine learning tasks” entails quite a broad meaning, so currently we focus on classification tasks of supervised machine learning.

We appreciate the comment of the reviewer concerning different datasets – so we have added four other gene expression cancer datasets (Liver, Lung, Prostate, Colorectal) with binary endpoint. We described these datasets in section 2.3.2 and included the results of Graph-CNN with different prior knowledge into the section 3.2.2. We have also compared the performance with the Random Forest method, which is commonly used on gene expression data. There is no substantial difference between methods on Lung, Prostate, and Colorectal cancer datasets. On the liver dataset Random Forest outperformed Graph-CNNs. 

The Embedding_net_v1 was compared with HPRD PPI, and two versions of randomized networks. The first version had the same topology as the Embedding_net_v1 but with permuted vertices. We permuted gene names over the vertices of Embedding_v1 to check the change in performance if the topology of the network is the same while the pair-wise interactions between proteins do not have a biological meaning. The second version was created as an artificial molecular network with random edges and random weights, described in the second paragraph of the section 2.4.5.

We have also added those lines into discussion:

“The difference in performance between Embedding_v1, HPRD PPI, and Embedding_v1 with permuted vertices was not substantial over the aforementioned datasets. We assume, that "small world properties" could be a reason for that - the majority of vertices’ pairs can be connected through the path with no more than 7 hops. 7-hops neighborhood were used by convolutional filters of all the Graph-CNNs we used. Also, for the lung and liver cancer, fully random network demonstrated similar performance. We hypothesize, that in the case of these datasets, the Graph-CNN was able to pick up patterns necessary for classification regardless of the vertex's connectivity or network information. A biological reason could stand behind this phenomenon. For instance, for lung and liver cancer data sets where more heterogeneous and expression correlations between genes did not coincide well with provided network topologies. We also noticed that the performance with the network with permuted vertices was always comparable. Only in prostate and colorectal cancer datasets does a random network with random weights worsens the classification performance. This fact is worth to be investigated further.”

As for the large breast cancer dataset, the performance comparison of the Graph-CNN with the State-of-the-Art methods is presented in the Table 1 of the paper Chereda, H., Bleckmann, A., Menck, K. et al. Explaining decisions of graph convolutional neural networks: patient-specific molecular subnetworks responsible for metastasis prediction in breast cancer. Genome Med 13, 42 (2021). https://doi.org/10.1186/s13073-021-00845-7

On the same dataset, we applied a Graph-CNN utilizing HPRD PPI, Random Forest without network data and “glmgraph” method implementing network-constrained sparse regression model using HPRD PPI network.

In another paper, Graph-CNN also performs better than Multilayer perceptron or Lasso logistic regression. We refer to Table 1 in paper:

Chereda H, Bleckmann A, Kramer F, Leha A, Beissbarth T. Utilizing molecular network information via graph convolutional neural networks to predict metastatic event in breast cancer. Stud Health Technol Inform. 2019; 267:181–6. https://doi.org/10.3233/SHTI190824

We would like to emphasize, that the performance of Graph-CNN compared to that of other machine learning methods is not a cornerstone of our paper. Here, we intended to use Graph-CNN as a validation tool for different underlying networks in the context of a given dataset. And so far, we observe, that only in the case of the breast cancer dataset, the Graph-CNN could demonstrate higher performance with Embedding_v1 network.

6. More figures about the methodology part will be helpful for reader to understand the method.

Unfortunately, it is not clear for us, which part of the methodology shall be explained better, but we assume this point corresponds to the application of the Graph-CNN. The schema of the prediction workflow can be found in [26, Figure 1 of]. We have added a reference to the Graph-CNN workflow figure in the subsection 2.5. Graph-Convolutional Neural Network.

Reviewer #2: In this manuscript, the authors introduced a text mining pipeline to produce word2vec embeddings for biological data. The proposed technique was evaluated on different tasks. The manuscript was well-written. All data and methods were introduced clearly.

I have the following comments to be addressed by the authors:

The performance of using Embedding_v1 is slightly better than Embedding_v2. The authors need to discuss the advantages of using the substitution of synonymous terms of genes, diseases, and drugs by their preferred terms.

Thank you for the suggestion. We have added the suggested content to the manuscript in the discussion section (line 558).

The authors gave one example in the discussion section in which using Embedding_v1 outperforms Embedding_v2. I recommend the authors providing additional examples.

Thank you for the recommendation. We have provided additional examples in the discussion section (615).

The main concern is comparing the performance of the proposed model with BioBERT in the tasks such as Reactome pathways, TRANSPATH pathways, GO biological processes, and Human disease ontology. The authors need to show the performance of BioBERT on these tasks.

We appreciate your suggestion. We have applied the same analysis with the same data resources with Bio-BERT embeddings and added the results to the section 3.1. We have also added the supplementary figure below (S8 Fig) that depicts the performance of Bio-BERT similarities. 

Did the authors test using other similarities rather than cosine similarity?

No, we didn’t use other similarities. Cosine similarity measures the cosine of the angle between two vectors. It is calculated using the inner (dot) product of the vectors and scaled by magnitude of each vector. In word embedding, it is more advantageous over magnitude-based metrics such as Euclidean distance since it focuses on the orientation of two vectors rather than their magnitudes and takes into account variations of occurrence counts between terms that are semantically similar. This makes the angle measure more resilient to variability of data and features' relative frequencies, whereas the magnitude of vectors is influenced by occurrence counts and heterogeneity of word neighbourhood. Any distance will be large when the vectors point different directions. Therefore, we used only cosine similarity since it is able to capture the similarity between two words even if they differ in frequency. Hence, if two vectors have different magnitudes and far apart by distance, they could still be oriented in the same direction.

How did the authors select the Similarity Threshold?

We have selected the similarity threshold 

1) to decrease the number of edges with low similarities by removing them

2) to make the number of vertices in a resulted set of genes mapped to the vertices of an embedding network comparable to that of HPRD PPI (6888). 

For example, in Embedding_net_v1 we deleted the edges that have weight less than 0.65. The edges were deleted, as well as some vertices. 

Reviewer #3: Review of manuscript “Text mining-based word representations for biomedical data analysis and machine learning tasks” by Alachram and co-workers.

I enjoyed reading this manuscript as it explained in some (but not too much) detail the process of using text mining with word2vec representation in biomedical text mining. The manuscript is logically structured, and I found most of my questions answered when reading further. However, I see at least two weaknesses with this work in its current state. Neither would disqualify the manuscript from publication, but at least this reviewer would like to see some more discussion on these limitations:

The improvement in the CNN classifier when based on the word2vec network over existing networks is very small, and not completely consistent. More worryingly or surprisingly perhaps, no network seems to perform significantly better than the random (unweighted) network. The differences seem to correspond to no more than one or two patients in the test set with 97 patients. No random weighted network (with random weights) was compared with. Why? It would also be interesting to see a comparison with other common (non-CNN) classifiers. Are those better or worse?

We agree that comparison of the of random network (with random weights) is an interesting point to consider. We have added a description of such a network in the second paragraph of 2.4.5 Random Network. 

We used the random network (with random weights) on four additional gene expression datasets (Liver, Lung, Prostate, and colorectal cancer) with binary endpoints. We described these datasets in section 2.3.2 and included the results of Graph-CNN with different prior knowledge into the section 3.2.2. The different prior knowledge included four different networks: Embedding_v1, HPRD PPI, Embedding_v1 with permuted vertices, and the random network (with random weights). We permuted gene names over the vertices of Embedding_v1 to check the change in performance if the topology of the network is the same while the pair-wise interactions between proteins do not have a biological meaning. We have also compared the performance with the Random Forest method, which is commonly used on gene expression data. 

There is no substantial difference between methods on Lung, Prostate, and Colorectal cancer datasets. On the liver dataset Random Forest outperformed Graph-CNNs.

We have also added those lines into discussion as a respond on the reviewer’s comment:

“The difference in performance between Embedding_v1, HPRD PPI, and Embedding_v1 with permuted vertices was not substantial over the aforementioned datasets. We assume, that "small world properties" could be a reason for that - the majority of vertice pairs can be connected through the path with no more than 7 hops. 7-hops neighborhood were used by convolutional filters of all the Graph-CNNs we used. Also, for the lung and liver cancer, fully random network demonstrated similar performance. We hypothesize, that in the case of these datasets, the Graph-CNN was able to pick up patterns necessary for classification regardless of the vertex's connectivity or network information. A biological reason could stand behind this phenomenon. For instance, for lung and liver cancer data sets where more heterogeneous and expression correlations between genes did not coincide well with provided network topologies. We also noticed that the performance with the network with permuted vertices was always comparable. Only in prostate and colorectal cancer datasets does a random network with random weights worsens the classification performance. This fact is worth to be investigated further.”

Concerning the breast cancer dataset, we agree the improvement does not seem to be very significant comparing HPRD PPI and Embedding_v1. Probably, our wording with 90% vs 10% data split in the section 3.2 (3.2.1 now) Graph-CNNs Results was confusing. We corrected it and added a couple of sentences in the section 3.2 (3.2.1 now) Graph-CNNs Results to emphasize that Table 1 shows the results from 10-fold cross validation. We have added it in the table’s title as well. Each metric in Table 1 is shown as its mean and standard error of the mean (st.e.) over 10 non-overlapping test folds. The performance difference would correspond to around 10 patients over whole dataset. 

Graph-CNN outperforms (10-fold cross validation) other common classifiers (Random Forest, Multilayer Perceptron, Lasso logistic regression) on the same dataset. The metrics value can be found in the Table 1 of these 2 papers

Chereda, H., Bleckmann, A., Menck, K. et al. Explaining decisions of graph convolutional neural networks: patient-specific molecular subnetworks responsible for metastasis prediction in breast cancer. Genome Med 13, 42 (2021). https://doi.org/10.1186/s13073-021-00845-7

Chereda H, Bleckmann A, Kramer F, Leha A, Beissbarth T. Utilizing molecular network information via graph convolutional neural networks to predict metastatic event in breast cancer. Stud Health Technol Inform. 2019; 267:181–6. https://doi.org/10.3233/SHTI190824

The networks are only compared on one breast cancer dataset. Though I admit we have also used this particular example, it contains some of the strongest gene-disease associations (and gene-gene with BRCA1 and BRCA2, and these with TP53) in the literature. It is therefore a bit of an extreme case. As the networks perform similarly, it would be interesting to see how they do on a wider range of examples.

This is a valid point, therefore we included four additional gene expression datasets (Liver, Lung, Prostate, and colorectal cancer) with binary endpoints. Please take a look at our answer on the previous point. 

Other comments:

Is the preprocessing step substituting synonymous terms by their preferred terms really trivial and error free? Quite a few gene and protein symbols/names are ambiguous, for example. Is the substitution global, or dependent on date of publication or context? What is the error rate of this substitution? For example, the official HGNC symbol for the serine dehydratase gene is “SDS”, but “SDS” in the literature more often refers to sodium dodecyl sulfate. This gene appears to have been removed from the networks, or at least I could not find it in the eBioMeCon services. The gene symbol for the amyloid beta precursor protein is “APP”, which also has another meaning, especially after lowercasing (though the list of nearest terms on the eBioMeCon looks reasonable for this gene). How often is an unambiguous term substituted for an ambiguous but preferred one in ‘Embedding_v1’? How often is an ambiguous term substituted for a non-synonymous one?

The substitution was global. This step is certainly not error free, and it is bit challenging to estimate the error rate from the resulting terms. It was applied during the pre-processing phase in which the context was not taken into account. We can’t prevent the risk of sematic ambiguity and both cases might definitely occur, unless the process was enhanced by entity recognition which would be another big task. However, the word context has surely played a major role during training along with the word frequency which would affect the neighbourhood of words. For instance, if an unambiguous term shares similar context with a preferred term, it is less likely for it to be substituted by an ambiguous term that has a different meaning. The same case for an ambiguous term substituted by a non-synonymous one. Unless one of the terms is more frequent and has different context, it is then more likely for both cases to happen and the context of the more frequent word would dominate. If the preferred term is more frequent, the synonymous term will disappear from the embedding and the context of the preferred term dominate and the other way around. One way to test the error rate from the resulting embedding is to check the word neighbours which would reveal whether those cases happened. In addition, one can also inspect the similarity values between the word and its nearest neighbours by comparing them to the corresponding values from the other embedding version. For example, if the similarity with the same neighbour was improved, this might suggest that a real synonymous term has substituted by a preferred term. Otherwise, (if the similarity was worsened) this might imply that one of the ambiguity cases has happened.

For the “SDS” gene, if a word does not exist in the resulting embedding, it means that the word was not learned by the model due to its frequency in text. During training, a certain threshold for the minimum times a word needs to be seen is selected. Therefore, if a word appeared less frequently in text (below the threshold), it will be dropped before training. 

Line 259: What is meant by “the number of proteins in the main connected component was also kept according to the comparable number of vertices in the HPRD PPI”? Did the authors fix the number of proteins in the main connected component to that in the existing PPT, and if so how? By removing edges below the “similarity threshold” as shown in Table 1?

Yes, it was understood correctly. We established the similarity threshold to make the number of vertices in the main connected component of the embedding networks comparable to that of HPRD PPI. The edges that are lower than the threshold were deleted, which also led to removal of some vertices. 

The values of the similarity threshold were around 0.65 for embedding networks. The edges with low similarities were removed to keep higher similarity. 

Line 295: How were these 10 terms chosen? Based on known strong associations? As mentioned above, BRCA1 is one of the more frequently appearing genes in the corpus, and having one of the strongest associations with a disease (breast cancer), so it is a bit of an extreme case (not saying that such are not useful or interesting, but it should at least be mentioned when introduced).

We chose these terms as they are of the most frequent terms used in literature and have strong biological associations with their neighbours in the embedding, so they can reflect how the text corpus size influences their relationships. In addition, this can also demonstrate how biologically meaningful are their relationships due to their strong similarities. We have edited the section 2.6 (line 353) in which we introduced the terms to make it more clear.

Table 1 - could not a random network, with random weights, be used for comparison? It would also be highly interesting to see the overlaps between the predictions. Are the misclassified cases more or less the same for all networks? The would definitely warrant a closer inspection of the gene expression dataset.

Yes, it is an interesting question. We have included a random network with random weights, and also tested it over 4 other cancer datasets (please see our previous comments above).

Concerning the overlaps of the predictions – we also think that it would be interesting to inspect the misclassified patients from the breast cancer dataset. For that we could explain the individual decisions of the Graph-CNN on misclassified patients utilizing methodology described in the paper: 

Chereda, H., Bleckmann, A., Menck, K. et al. Explaining decisions of graph convolutional neural networks: patient-specific molecular subnetworks responsible for metastasis prediction in breast cancer. Genome Med 13, 42 (2021). https://doi.org/10.1186/s13073-021-00845-7

We thought about this possibility, but we decided this question could be addressed in future research since currently, it would drastically shift the scope of this paper from Natural Language Processing towards the application of the methodology described in the paper above.

Line 474: should be brca2, not “brac2” (different gene, not strongly associated).

Thank you for noting this. It was a spelling error. We have corrected it.

Sincerely, 

Halima Alachram and Hryhorii Chereda on behalf of all authors.

---

## [Decision Letter · Decision Letter 1]

4 Oct 2021

Text mining-based word representations for biomedical data analysis and protein-protein interaction networks in machine learning tasks

PONE-D-20-38223R1

Dear Dr. Alachram,

We’re pleased to inform you that your manuscript has been judged scientifically suitable for publication and will be formally accepted for publication once it meets all outstanding technical requirements.

Kind regards,

Khanh N.Q. Le

Academic Editor

PLOS ONE

Additional Editor Comments (optional):

Reviewers' comments:

Reviewer's Responses to Questions

**Comments to the Author**

1. If the authors have adequately addressed your comments raised in a previous round of review and you feel that this manuscript is now acceptable for publication, you may indicate that here to bypass the “Comments to the Author” section, enter your conflict of interest statement in the “Confidential to Editor” section, and submit your "Accept" recommendation.

Reviewer #2: All comments have been addressed

Reviewer #3: All comments have been addressed

2. Is the manuscript technically sound, and do the data support the conclusions?

Reviewer #2: Yes

Reviewer #3: Yes

3. Has the statistical analysis been performed appropriately and rigorously? 

Reviewer #2: Yes

Reviewer #3: Yes

4. Have the authors made all data underlying the findings in their manuscript fully available?

Reviewer #2: Yes

Reviewer #3: Yes

5. Is the manuscript presented in an intelligible fashion and written in standard English?

Reviewer #2: Yes

Reviewer #3: Yes

6. Review Comments to the Author

Reviewer #2: (No Response)

Reviewer #3: The authors have address my previous concerns or suggestions (often while also addressing those of the other two reviewers). I recommend acceptance of this manuscript.

Minor suggestion:

lines 381 and 410: change "didn’t" to "did not"

7. PLOS authors have the option to publish the peer review history of their article (what does this mean?). If published, this will include your full peer review and any attached files.

Reviewer #2: No

Reviewer #3: No

---

## [Editor Report · Acceptance letter]

8 Oct 2021

PONE-D-20-38223R1 

Text mining-based word representations for biomedical data analysis and protein-protein interaction networks in machine learning tasks 

Dear Dr. Alachram:

I'm pleased to inform you that your manuscript has been deemed suitable for publication in PLOS ONE. Congratulations! Your manuscript is now with our production department. 

Kind regards, 

on behalf of

Dr. Khanh N.Q. Le 

Academic Editor

PLOS ONE